# FASTER AND ACCURATE NEURAL NETWORKS WITH SEMANTIC INFERENCE

## ABSTRACT

Deep neural networks (DNNs) usually come with a significant computational and data labeling burden. While approaches such as structured pruning and mobile-specific DNNs have been proposed, they incur in drastic accuracy loss. Different from prior work, in this paper we leverage the redundancy present in latent representations to drastically reduce the computational load with unnoticeable performance loss. Specifically, we show that semantically similar inputs share a significant number of filter activations, especially in the earlier layers. Therefore, semantically similar classes can be "clustered" so as to create cluster-specific subgraphs. These may be "turned on" when an input belonging to a semantic cluster is being presented to the DNN, while the rest of the DNN can be "turned off". To this end, we propose a new framework called *Semantic Inference* (SINF). In short, SINF (i) uses a small auxiliary classifier to identify the semantic cluster of the input object; and then (ii) executes the corresponding subgraph extracted from the base DNN to obtain the final prediction. To extract each cluster-specific subgraph, we propose a new approach named *Discriminative Capability Score* (DCS) that effectively finds such subgraph capable of discriminating among the members of that specific semantic cluster. Importantly, DCS is independent from SINF, as it is a general-purpose quantity that can be calculated for any DNN. We benchmark the performance of DCS on VGG16, VGG19, and ResNet50 DNNs trained on the CIFAR100 dataset against 6 state-of-the-art pruning approaches. Our results show that (i) SINF reduces the inference time of VGG19, VGG16, and ResNet50 respectively by up to 35%, 29% and 15% with only 0.17%, 3.75%, and 6.75% accuracy loss; (ii) DCS achieves respectively up to 3.65%, 4.25%, and 2.36% better accuracy with VGG16, VGG19, and ResNet50 with respect to existing discriminative scores; (iii) when used as a pruning criterion, DCS achieves up to 8.13% accuracy gain with 5.82% less parameters than the existing state-of-the-art published at ICLR 2023; (iv) when considering per-cluster accuracy, SINF performs on average 5.73%, 8.38% and 6.36% better than the original VGG16, VGG19, and ResNet50. We share our code for reproducibility.

## 1 INTRODUCTION

Deep neural networks (DNNs) have produced significant advances in computer vision (CV), (Krizhevsky et al., 2012; Kirillov et al., 2023; Redmon et al., 2016), natural language processing (NLP) [Vaswani et al. (2017)], and multi-modal tasks (Radford et al., 2021), just to name a few. Usually, DNNs process a very large number of parameters. For example, the state of the art YoLov8 uses a DNN backbone with 53 layers and 40M parameters (Terven & Cordova-Esparza, 2023). On the other hand, DNNs are increasingly being used in resource-constrained mobile systems. For example, unmanned autonomous vehicles (UAVs) and self-driving cars need to frequently perform object detection (Wu et al., 2020) and semantic segmentation (Mo et al., 2022) to avoid obstacles during navigation and build detailed 3D maps (Wang et al., 2020; Fraga-Lamas et al., 2019; Riti Dass (Medium), 2018). Applying state-of-the-art DNNs in these scenarios is hardly feasible for real-time edge deployment.

As discussed in Section 2, a plethora of existing work has been devoted to reduce the complexity of DNNs (Tan & Le, 2019; Howard et al., 2017; Iandola et al., 2016; Sandler et al., 2018). Mobile-specific DNNs such as MobileNet (Sandler et al., 2018) and MnasNet (Tan et al., 2019) decrease

the computational requirements at the detriment of accuracy. For example, MobileNet loses up to 6.4% in accuracy compared to ResNet-152 (He et al., 2016). Alternative approaches include pruning (Han et al., 2015b; Li et al., 2018; Chen et al., 2019; Tanno et al., 2019; Singh et al., 2019; Kaya et al., 2019; Yao et al., 2017; Huang et al., 2020), quantization (Han et al., 2015a; Qin et al., 2022; Cai et al., 2020), and coding (Gajjala et al., 2020; Han et al., 2015a), which also incur in excessive DNN performance loss. Another key issue is that existing approaches do not guarantee faster inference as previous methods depend on the way the DNN is implemented in hardware. For example, unstructured pruning does not lead to faster inference in practice since the majority of the DNN circuitry has to be executed regardless of the application (Wen et al., 2016; Ma et al., 2022). On the other hand, structured pruning, which encompasses both filter and layer pruning, makes the inference faster and less energy-expensive (Ma et al., 2022).

In stark opposition to prior work, in this paper, we propose *Semantic Inference* (SINF) to reduce the number of weights for a DNN implemented on an edge device. Our key intuition is that in practical mobile settings, DNN inputs usually originate from a small subset of training classes and are *semantically similar* and *highly correlated* over time. For example, DNNs deployed in drone-based surveillance systems may only need to detect and identify certain classes (e.g., people, cars, animals) and will hardly encounter indoor objects. Another intuition – proven in Figure 1 – is that semantically similar inputs share a significant number of filter activations compared to semantically dissimilar inputs, especially in the earlier layers. As an intuitive example, images of seals share significantly more filter activations with images of dolphins than with images of tables. *We use this to logically rearrange the DNN so that only a portion of the DNN corresponding to the semantic class of the current input will be executed.*

This paper makes the following novel contributions:

• We propose a new inference framework called SINF, which logically partitions the DNN into subgraphs considering semantic similarities among different classes, so that only the subgraph relevant to the input's semantic cluster gets activated. To this end, we propose Discriminative Capability Score (DCS) to find the filters that can best distinguish semantically similar classes;

• We benchmark the performance of our SINF on VGG16, VGG19, and ResNet50 DNNs trained on the CIFAR100 dataset. We compare DCS against state-of-the-art discriminative algorithms proposed by Mittal et al. (2019), Molchanov et al. (2019), Hu et al. (2016), Sui et al. (2021), and Lin et al. (2020). We also use DCS as a pruning approach and compare it against the work by Murti et al. (2023), which like our work does not require retraining and/or fine-tuning. Our results show that (i) SINF reduces the inference time of VGG19, VGG16, and ResNet50 respectively by up to 35%, 29% and 15% with only 0.17%, 3.75%, and 6.75% accuracy loss; (ii) DCS achieves respectively up to 3.65%, 4.25%, and 2.36% better accuracy on VGG16, VGG19, and ResNet50 compared to existing discriminative scores; (iii) when used as a pruning criterion, DCS achieves up to 8.13% accuracy gain with 5.82% less parameters than the existing state of the art published at ICLR 2023; (iv) when considering per-cluster accuracy, SINF performs on average 5.73%, 8.38% and 6.36% better than the original VGG16, VGG19, and ResNet50.

## 2 RELATED WORK

**Model Pruning**: The *lottery ticket hypothesis* introduced by **?** has spurred a plethora of research work in DNN pruning. Even before that, Han et al. (2015b) proposed weight-norm-based unstructured pruning, while Paul et al. (2023) tried to explain the success of the magnitude-based pruning methods. Li et al. (2017) used the $L_1$ norm of the kernel weights to prune entire filters. However, weight-norm-based strategies do not directly take into account the importance of the filters or parameters to preserve the DNN accuracy. Another approach is first-order gradient based (Molchanov et al., 2017; 2019) which estimate the importance of the filters based on the gradient of the loss function. Another class of techniques leverage the filter activation maps. For example, RoyChowdhury et al. (2017) study the presence of duplicate neurons and determines that convolutional layers are prone to developing redundant duplicate filters. To find such filters, Sui et al. (2021) uses the change in nuclear norm of the matrix formed from the activation maps when individual filters are removed from a layer. Lin et al. (2020) use the expected rank of the feature maps. Chen et al. (2023) explain the soft-threshold pruning as an implicit case of Iterative Shrinkage-Thresholding. *Although these methods determine the redundant filters, they fail to focus on the filters which are necessary to distinguish among the classes. Moreover, all of these methods require fine-tuning after pruning.* When the fine-tuning is not possible, these methods do not provide satisfactory performance. Re-

cently, Murti et al. (2023) propose a retrain-free *IterTVSPrune* approach based on Total Variational Distance (TVD) (Verdú, 2014). Here, we take a semantics-based approach and attempt to find filters able to best discriminate classes belonging to a given semantic cluster.

**Quantization and Coding**: The seminal work by Han et al. (2015a) compressed the DNN through quantization and Huffman coding to reduce the memory footprint. Among more recent work, post-training quantization [Cai et al. (2020), Fang et al. (2020)] and quantization-aware training [Liu et al. (2021), Bhalgat et al. (2020), Zhong et al. (2022)] have been proposed. Qin et al. (2022) pushes the boundary using single-bit quantization of the popular language model Bidirectional Encoder Representations from Transformers(BERT). Li et al. (2020) designed a layer-wise symmetric quantizer with the learnable clip value only for high-level feature extraction module. Tu et al. (2023) recently designed an algorithm for network quantization catered to the needs of image super resolution. Gajjala et al. (2020) proposes three variants of Huffman encoding to compress the gradients for distributed training of neural networks. Both quantization and coding are complementary to the SINF and can be used to achieve further improvement in performance.

**Early Exit in Neural Network** Early exit was proposed by Teerapittayanon et al. to make the DNN inference dynamic by using auxiliary (and relatively small) neural networks attached to the output of the DNN layers. Based on the confidence of the prediction of the auxiliary networks, the decision to traverse the remaining layers is made [Matsubara et al. (2021)]. The training of the auxiliary classifiers can be done jointly with the backbone network as done by Elbayad et al. (2020), Zhou et al. (2020), and Pomponi et al. (2022). The classifiers can be trained using either cross-entropy loss [Lo et al. (2017), Wang et al. (2019)], or through knowledge distillation [Phuong & Lampert (2019), Li et al. (2019)]. The classifiers can also be trained separately as in Liu et al. (2020), Garg & Moschitti (2021), and Xin et al. (2020). Han et al. (2023) tries to improve the performance of the early classifiers by using block-dependent loss which uses information from a subset of the exits close to a block to train it. Dong et al. (2022) addresses the wasteful computation of the early auxiliary classifiers when they are not confident enough by predicting which early exit to use using a lightweight "Exit Predictor". Narayan et al. (2023) modeled the exit selection as an online learning problem and proposed to choose the exit in an unsupervised way. SINF uses an auxiliary classifier but the end goal is not early inference but to predict the next path to route the sample input to.

## 3 DIVIDING A DNN INTO SEMANTIC SUBGRAPHS

Let $\mathcal{D}$ be a labeled dataset with classes taken from a set $\mathcal{K}$. We define a set of $K$ clusters of classes $\{\gamma_1, \cdots, \gamma_K\}$, such that $\gamma_1 \cup \gamma_2 \cup \cdots \gamma_K = \mathcal{K}$. We assume that these clusters are defined based on application-level similarities (e.g., classes related to flowers, insects, etc.) or pre-defined at the dataset level (e.g., as in the CIFAR100 dataset). We define $\mathcal{F}$ as a DNN trained on dataset $\mathcal{D}$. By viewing $\mathcal{F}$ as a computation graph, we define the Semantic DNN Subgraph Problem (SDSP).

---

**Semantic DNN Subgraph Problem (SDSP)**

Find $K$ proper subgraphs $\mathcal{F}_{\gamma_i} \cdots \mathcal{F}_{\gamma_K}$ such that

$$\mathcal{B}_{eval}(\mathcal{F}, \mathcal{D}) \leq \frac{1}{K} \sum_{i=1}^{K} \mathcal{B}_{eval}(\mathcal{F}_{\gamma_i}, \mathcal{D}_{\gamma_i}), \quad (1)$$

where $\mathcal{F}_{\gamma_i} \subset \mathcal{F}$ and $\mathcal{D}_{\gamma_i} \subset \mathcal{D}$ are respectively the proper subgraphs of $\mathcal{F}$ and subset of the dataset corresponding to the partition $\gamma_i$. The function $\mathcal{B}_{eval}$ is the evaluation metric being used to score $\mathcal{F}$ on dataset $\mathcal{D}$ as well as to score the subgraphs on their corresponding semantic clusters. A higher value of $\mathcal{B}_{eval}$ is assumed to correspond to better performance.

---

In other words, the subgraph $\mathcal{F}_{\gamma_i}$ contains the nodes of $\mathcal{F}$ which best classifies the members of the corresponding partition $\gamma_i$ on $\mathcal{D}_{\gamma_i}$. We choose the evaluation metric $\mathcal{B}_{eval}$ as accuracy in this work but it can be set to any other performance metrics as needed. If a metric doesn't satisfy the assumption that higher value corresponds to better performance, it cans till be used simply by taking the negative of the metric. We perform a series of experiments to validate the intuition behind the SDSP. It is well known that DNN filters identify parts of objects, colors or concepts. These filters are shared among classes to reduce the number of parameters of the DNN [Bau et al. (2017)]. On the other hand, filter activations become sparser as the DNN becomes deeper, with filters reacting only to specific inputs belonging to specific classes. This phenomenon can be observed in the top portion of Figure 1, which shows the average filter activation strength for the "otter" and "seal" classes in the the 40th and 49th convolutional layers of ResNet50 trained on CIFAR100.

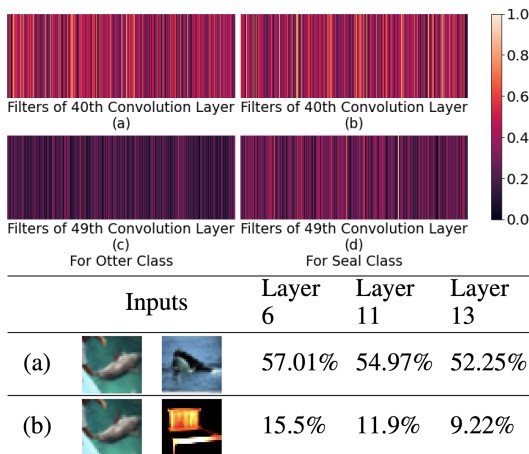

| Inputs | | Layer 6 | Layer 11 | Layer 13 |
|---|---|---|---|---|
| (a) | | 57.01% | 54.97% | 52.25% |
| (b) | | 15.5% | 11.9% | 9.22% |

Figure 1: (top) Filter activations in ResNet50; (bottom) Percentage of filters shared between (a) semantically similar classes – "dolphin" and "whale"; (b) semantically dissimilar classes – "dolphin" and "table".

To obtain these results, we have taken the average of each filter activation vector for each input in the training set corresponding to the two classes, and performed min-max normalization to obtain values between 0 and 1. This experiment reinforces the notion that filters in earlier layers are less specialized than filters in deeper layers. Moreover, it remarks that filters from semantically similar classes get similarly activated, especially in earlier layers. *To put it in more quantitative terms, the $L_1$ distance of the activation maps of the mentioned classes in the $40^{th}$ layer is 0.028, while the same for the $49^{th}$ layer is 0.111.* To further investigate this critical aspect, we have performed additional experiments where we have computed the percentage of filters "shared" among different classes for each layer of VGG16. Specifically, we have tagged each filter with the top 20 classes for which it gets activated. For each pair of classes, their similarity is calculated as the number of filters tagged with both classes over the number of filters tagged with at least one of the classes. The results are shown in the bottom portion of Figure 1, where the first row shows the filters shared between the "dolphin" and "whale" classes – two semantically similar classes. The second row shows the filter sharing between two semantically dissimilar classes - "dolphin" and "table". *As the numbers suggest, the semantically similar classes share more filters than semantically dissimilar classes.* These results further confirm that filter sharing among classes decreases as we go deeper in the DNN.

## 4 DISCRIMINATIVE CAPABILITY SCORE

We describe the procedure to obtain the DCS for a given layer $l$ and given semantic cluster $\gamma_m$ in Algorithm 1. We start by considering a generic layer $l$ of a DNN with $C_{out}^l$ number of output channels. We define the activation output of channel $c_i$ of this layer for an input sample $X^j$ as $\mathbf{A}_{l,c_i}^j$, $\mathbf{A}_{l,c_i}^j \in \mathbb{R}^{C_{out}^l \times H \times W}$ where $H$ and $W$ are respectively the height and width of the activation output. For a specific input $X^j$, from line 5-10 of algorithm 1, the following procedures are performed. For each output channel, we obtain the corresponding activation map $\mathbf{A}_{l,c_i}^j$. Then we perform an adaptive pooling operation $\mathcal{P}(\cdot)$ to reduce the size of the feature map of each channel from $H \times W$ to $k \times k$. Flattening the feature map gives a feature vector $\mathbf{F}_{l,c_i}^j$ of size $k^2$. Next, the feature vectors obtained from all the filters are concatenated to generate the complete feature vector $\mathbf{F}_l^j$, $\mathbf{F}_l^j \in \mathbb{R}^{N_f}$ for layer $l$ where $N_f = C_{out}^l \cdot k^2$ is the length of the feature vector. In line 12 of algorithm 1 the $\{\mathbf{F}_l^j, t^j\}_{j=1}^{j=|\mathcal{D}_{\gamma_m}|}$ thus obtained is used to find a transformation $\mathbf{W} \in \mathbb{R}^{N_c \times N_f}$ ($N_c$ = Number of classes in semantic cluster $\gamma_m$) that optimizes objective function $\mathcal{L}_{DOF}$ as shown in Equation 2:

$$\mathbf{W}_l^* = \underset{\mathbf{W}}{\arg\min} \frac{1}{|\mathcal{D}_{\gamma_m}|} \sum_{j=1}^{j=|\mathcal{D}_{\gamma_m}|} \mathcal{L}_{DOF}(\mathbf{W} \cdot \mathbf{F}_l^j, t^j), \tag{2}$$

where $t^j$ denotes the target value or label for the input sample $X^j$ in the dataset $\mathcal{D}_{\gamma_m}$ corresponding to a class cluster $\gamma_m$. $\mathbf{W}_l^*[m, f_i]$ connects the $f_i$-th feature in a feature vector to the the $m$-th class. As such, each column of $\mathbf{W}_l^*$ correspond to the contribution of each feature to differentiate among the classes. The same goes for the elements in feature importance matrix (line 13) $\mathbf{I}^l = \mathbf{W}_l^* \odot \nabla_{\mathbf{W}_l^*} \bar{\mathcal{L}}_{DOF}$. Here, $\bar{\mathcal{L}}_{DOF}$ is the objective function averaged over the samples. The multiplication with the gradient lets us consider not only the strength but also the sensitivity of the objective function $\mathcal{L}_{DOF}$ to the features. We use $\mathbf{I}^l$ to define our metric DCS. We define $s_i$ as the norm of the $i$-th column of the feature importance matrix $\mathbf{I}^l$. $s_i$ represents the score contribution of a feature $f_i$ in discriminating among the semantically similar classes. Taking norm of the columns of $\mathbf{I}^l$ converts the matrix $\mathbf{I}^l$ into a vector $\mathbf{s}$ of length $C_{out}^l \cdot k^2$. The final step is to attribute the DCS to individual filters. (Line 17) We use a group norm operation $\mathcal{G}(\cdot)$ on the obtained feature score vector $\mathbf{s}_l$ where the feature scores are grouped into vectors $\mathbf{u}_{c_i}$ consisting of $k^2$ consecutive feature score values corresponding to channel $c_i$. Then DCS of channel $c_i$ of layer $l$ is obtained as follows:

---

**Algorithm 1** Computing the DCS for Filters in Layer $l$ and Semantic Cluster $\gamma_m$

---

**Input**:        Dataset for given semantic cluster $\gamma_m$: $\mathcal{D}_{\gamma_m} = \{X^j, t^j\}_{j=1}^{|\mathcal{D}_{\gamma_m}|}$
                   Pretrained DNN: $\mathcal{F} = \mathcal{F}_{L-1} \circ \mathcal{F}_{L-2} \circ \ldots \circ \mathcal{F}_0$
                   Discriminative Objective Function: $\mathcal{L}_{DOF}$

**Output**:      Discriminative Capability Score of filters of $l^{\text{th}}$ layer = $\text{DCS}_l$

---

1:  $\mathbf{s}_l$ = Empty list ()                            // To store the contribution of the features
2:  **for** $X^j, t^j$ in $\mathcal{D}_{\gamma_m}$ **do**
3:        $\mathbf{F}_l^j$ = Empty list ()                        // To store the feature values of individual samples
4:        $C_{out}^l \leftarrow$ Number of filters in $l$-th layer
5:        **for** $c_i = 0, 1, 2, \ldots, C_{out}^l - 1$ **do**
6:            $\mathbf{A}_{l,c_i}^j \leftarrow \mathcal{F}_l \circ \mathcal{F}_{l-1} \circ \ldots \circ \mathcal{F}_0(X^j)$
7:           $\tilde{\mathbf{A}}_{l,c_i}^j \leftarrow \mathcal{P}(\mathbf{A}_{l,c_i})$            //Adaptive Pooling Function with output shape $k \times k$
8:           $\mathbf{F}_{l,c_i}^j \leftarrow \text{Flatten}(\tilde{\mathbf{A}}_{l,c_i}^j)$
9:           $\mathbf{F}_l^j \leftarrow \mathbf{F}_l^j \sim \mathbf{F}_{l,c_i}^j$             // $\sim$ denotes concatenation operation
10:     **end for**
11: **end for**
12: $\mathbf{W}_l^* \leftarrow \underset{\mathbf{W}}{\text{argmin}} \ \frac{1}{|\mathcal{D}_{\gamma_m}|} \sum_{j=1}^{j=|\mathcal{D}_{\gamma_m}|} \mathcal{L}_{DOF}(\mathbf{W} \cdot \mathbf{F}_l^j, t^j)$ //$\mathbf{W}$ is a transformation matrix
13: $\mathbf{I}^l \leftarrow \mathbf{W}_l^* \odot \nabla_{\mathbf{W}_l^*} \bar{\mathcal{L}}_{DOF}$      //$\odot$ and $\bar{\mathcal{L}}_{DOF}$ denote elementwise multiplication and average loss
14: **for** $i = 0, 1, 2, \ldots C_{out}^l - 1$ **do**
15:        $\mathbf{s}_l \leftarrow \mathbf{s}_l \sim \|\mathbf{I}_{:,i}^l\|_2$ //$(.)_{:,i}^*$ denotes the weight column corresponding to the i th feature
16: **end for**
17: $\text{DCS}_l \leftarrow \mathcal{G}(\mathbf{s}_l)$                           // $\mathcal{G}$ = Group Norm Operation

---

$$\text{DCS}_{c_i} = \sqrt{\sum_j u_{c_i,j}^2} \tag{3}$$

where $\text{DCS}_{c_i}$ is the desired discriminative capability score of the channel $c_i$ and $u_{c_i,j}$ is the $j$ th element of the feature cluster corresponding to the same filter. We use this DCS to extract the sub-graph for the semantic clusters.

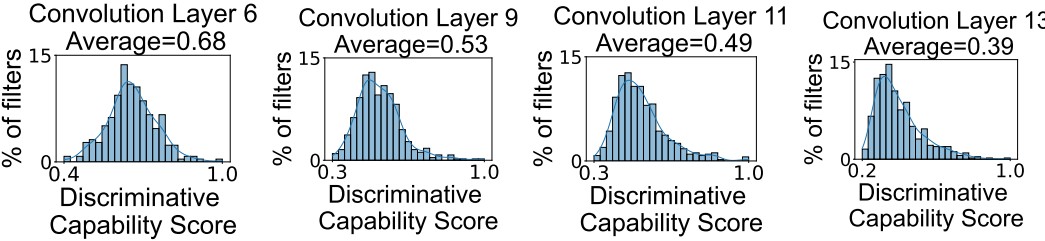

Figure 2: The DCS distribution for cluster "fish" of CIFAR100 in different layers of VGG16.

Figure 2 shows the DCS distributions obtained in layer 6, 9, 11, and 13 of VGG16 by considering the cluster "fish" of CIFAR100. Figure 2 confirms that deeper layers are more specialized for individual classes, and thus the average DCS for the filters in the deeper layers is smaller – 0.68 for layer 6 vs 0.39 for layer 13. This aligns with the observation of the previous section - *the deeper one goes into a deep neural network, the less number of filters are needed for discrimination among the classes.* So, we can say DCS captures the filter activation pattern of the DNN.

## 5  SEMANTIC INFERENCE (SINF)

A step-by-step overview of the main operations of SINF is summarized in the top portion of Figure 3. Prior to the deployment of the DNN, SINF uses the DCS to construct the semantic subgraphs, as explained later. After deployment, upon receiving an input, SINF first classifies which semantic cluster the input belongs to (**step 1**). To this end, a *Common Feature Extractor* is trained to extract the features to correctly predict the semantic cluster (**step 2**). This is actually performed by the *Semantic*

*Route Predictor* (SRP), whose structure is detailed later in this section (**step 3**). Based on the SRP output, the input will be routed to the selected semantic subgraph by using a *Feature Router* (**step 4**).

Finally, the inference output is obtained from the appropriate subgraph (**step 5**). We remark that although we are representing each subgraph separately for better graphical clarity, in practice the separation is only from a logical perspective. In other words, no additional memory beyond the annotations needed to characterize each subgraph is used by SINF.

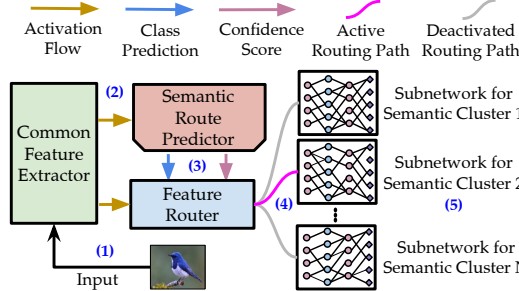

Figure 3: Overview of Semantic Inference (SINF).

**Semantic Route Predictor.** The purpose of the semantic route predictor is to predict the semantic clusters an input sample belongs to so that it can be forwarded towards the corresponding route for final prediction. An auxiliary classifier $\chi$, attached after the $M-1$-th layer of the base model $\mathcal{F}$ is used for this task. The earliest layer which provides good prediction (an accuracy threshold of 75% is used in this case for the sake of experimentation but this depends on the application-level constraints) for semantic routes is chosen as the $M-1$ th layer. As we decide on the $M-1$-th layer, the part of the base model up to this layer becomes the *Common Feature Extractor*. The architecture of the auxiliary classifier consists of two convolution layers, followed by an adaptive average pooling layer stacked on top of three fully connected layers. SINF uses the convolution layers to tailor the activation map from layer $l$ of the base model for classification of the semantic clusters. To train the auxiliary classifier $\chi$, the section of the base model up to the $M-1$-th layer of $\mathcal{F}$ is frozen and the classifier is trained in supervised fashion using $\{\mathbf{A}_{M-1}^{j}, \gamma_m^j\}_{j=1}^{j=|\mathcal{D}|}$ as the dataset. Here, $\mathbf{A}_{M-1}^{j}$, and $\gamma_m^j$ are respectively the activation of the $M-1$-th layer of the base model and the ground truth semantic cluster for the $j$-th sample. As we are considering a pre-trained base network, we train the auxiliary classifier separately from the base network using the activations obtained from the $M-1$-th layer. The output of semantic route predictor is the probability distribution over the $K$ different semantic clusters and the input of the base network is predicted to belong to the cluster with the highest probability. In our case, the value of $K$ is 20 as we have 20 semantic clusters.

**Extraction of Subgraphs.** The extraction of the subgraph follows the procedure described in Algorithm 2. We define $L$ and $M$ as respectively the last layer of the base model and the layer just after the Common Feature Extractor. We define $r_l$ as the percentage of retained filters in generic layer $l$. For semantic cluster $\gamma_i$, we iterate from layer $L$ to layer $M$ to extract the subgraph. For each layer $M \leq l \leq L$, we calculate $r_l(r_L \leq r_l \leq r_M)$ (Line 5), as well as the DCS score (Line 6) of the filters using Algorithm 1. We rank the filters based on the DCS score and the indices of the top $r_l$ percent filters are saved. This is repeated for all the semantic clusters. If the average accuracy of the extracted subgraphs for the semantic clusters is above an accuracy threshold $\tau_{acc}$, the indices of the filters belonging to the subgraphs are stored. The accuracy threshold $\tau_{acc}$ is set to the accuracy of the original DNN as shown in Equation 1. This procedure is performed for different values of $r_L$ and $r_M$. In this work, $r_L$ is set between 90% and 10%, with steps of 10, while $r_M$ is set between 10% and 1%, with steps of 2. In this way we are performing a search by varying the number of retained filters in different layers. allowing us to find multiple sub-graphs satisfying our constraint as set in 1. Based on the application level performance constraint, we can then choose the optimum model based on additional requirements (e.g. sub-graph size, latency). Motivated by the observation that the deeper we go into the neural network, the less number of filters we need to represent the classes, we linearly decrease the percentage of filters retained from layer $M$ to layer $L$ of the DNN according to equation 5 presented in Line 5 of algorithm 2. Finally, Although we have used categorical cross entropy for $\mathcal{L}_{DOF}$, this can be set to any loss function.

**Feature Router.** The DCS decision can be improved if SINF conditions the output of the semantic router predictor $\chi$ on its confidence. The feature router calculates this confidence and routes the activation maps to the appropriate specialized path. It takes the activation map from the semantic router predictor $\chi$ along with the probability distribution from its prediction layer. To compute the confidence of the classifier on individual decisions, the feature router employs the lightweight metric proposed by Park et al. (2015). The confidence score can be calculated as $C_\chi = P_h - P_{sh}$, using the highest ($P_h$) and the second highest probabilities ($P_{sh}$) for individual semantic clusters. The confidence score is a proxy for the probability that the inference aligns with the correct label.

The confidence score of a classifier correlates with the probability of correct inference as is shown by the Park et al. (2015). If the confidence score exceeds a threshold, the activation map is routed to the subgraph corresponding to the predicted semantic cluster - in other words, only the filters corresponding to this subgraph are turned on. Otherwise the base model is used for final decision.

---

**Algorithm 2** Subgraph Extraction for Semantic Clusters

---

**Input**:        Partitioned Dataset $\mathcal{D} = \{\mathcal{D}_{\gamma_1}, \mathcal{D}_{\gamma_2} \ldots \ldots \mathcal{D}_{\gamma_K}\}$
                      Pretrained DNN: $\mathcal{F} = \mathcal{F}_{L-1} \circ \mathcal{F}_{L-2} \circ \ldots \circ \mathcal{F}_0$
                      Discriminative Objective Function: $\mathcal{L}_{DOF}$
                      Filter Retention Percentage at layer $L = r_L$
                      Filter Retention Percentage at layer $M = r_M$
                      Accuracy Threshold = $\tau_{acc}$

**Output**:      Filter annotations for the subgraphs for semantic clusters = **SA**[]

1:  **SA** = Empty dictionary()               // To store the final filters annotations for extracted subgraphs
2:  **for** $\mathcal{D}_{\gamma_i}$ in $\mathcal{D}$ **do**
3:      **SA**$_{\gamma_i}$ = Empty dictionary()
4:      **for** $l = L, L-1, \ldots, M$ **do**
5:         $r_l \leftarrow r_M + \frac{(l-M)(r_L-r_M)}{L-M}$           // Percentage of filters to retain at layer $l$
6:         DCS$_l \leftarrow DCS(\mathcal{F}, \mathcal{D}_{\gamma_i}, \mathcal{L}_{DOF})$       //Obtaining DCS Scores from algorithm 1
7:         Rank the filters
8:         Save the indices of top $r_l$ percent filters in **SA**$_{\gamma_i}$ with layer number as key
9:      **end for**
10:     Calculate $acc_{avg} = \frac{1}{K} \sum_{i=1}^{i=K} (accuracy(\mathcal{F}_{\gamma_i}))$
11:     **if** $acc_{avg} \geq \tau_{acc}$ **then**
12:         Save **SA**$_{\gamma_i}$ in **SA** with $\gamma_i$ as key
13:     **end if**
14: **end for**

---

## 6   Experimental Evaluation

To quantify improvement of SINF, we consider existing pruning approaches by Molchanov et al. (2019), Hu et al. (2016), Mittal et al. (2019), Sui et al. (2021), and Lin et al. (2020). In addition, to compare with a pruning approach that does not require retraining, we also consider the work Murti et al. (2023) published at ICLR 2023. We chose the CIFAR100 dataset for our experiments, since its 100 classes are pre-grouped into 20 semantically similar classes representing our semantic clusters.

**Impact of Confidence Threshold.** We first evaluate the impact of the confidence threshold $\alpha$, given its importance in achieving the right balance between latency and accuracy in SINF. The top row of Figure 4 shows the cumulative distribution of the confidence values of SINF for the VGG16, VGG19, and ResNet50 DNNs computed over the test set. In addition, the bottom portion of Figure 4 shows the decrease in accuracy and the relative latency with respect to the original DNN as a function of $\alpha$. As expected, increasing $\alpha$ increases the accuracy while also decreasing the gain in latency. As such, the confidence threshold $\alpha$ acts as a hyper parameter to find the needed trade-off between accuracy and latency. We notice that with VGG19, the overall accuracy actually increases by up to 0.49% for $0.4 \leq \alpha \leq 0.9$. Looking at the confidence distribution of VGG19, we can see that the auxiliary classifier is more confident about its predictions compared to the case of VGG16. In the best case, SINF reduces the inference time by up to 35%, 29% and 15% with only 0.17%, 3.75%, and 6.75% accuracy loss for VGG19, VGG16, and ResNet50 respectively.

**DCS vs Existing Discriminative Metrics.** To evaluate the effectiveness of DCS with respect to prior approaches, we use discriminative metrics proposed in existing work while keeping the same inference structure of SINF. Figure 5 compares DCS against gradient-based approaches *Sensitivity* by Mittal et al. (2019) and *Taylor* by Molchanov et al. (2019), sparsity of activation based approach *APOZ* by Hu et al. (2016), channel-independence based approach *CHIP* by Sui et al. (2021), and an approach based on channel importance named *HRANK* by Lin et al. (2020). All the approaches are compared without retraining the DNN, as proposed in SINF. Figure 5 shows that in the best case, DCS has 15% higher accuracy than the second-best approach *Taylor* for VGG16 with 75% sparsity (i.e., percentage of parameters dropped). For VGG19, DCS achieves in the best case 6.54% higher accuracy than the second-best approach *Taylor* at 63% sparsity. Lastly, in the case of ResNet50, the best case is attained at 51% sparsity, where DCS presents 14.87% more accuracy than the second-

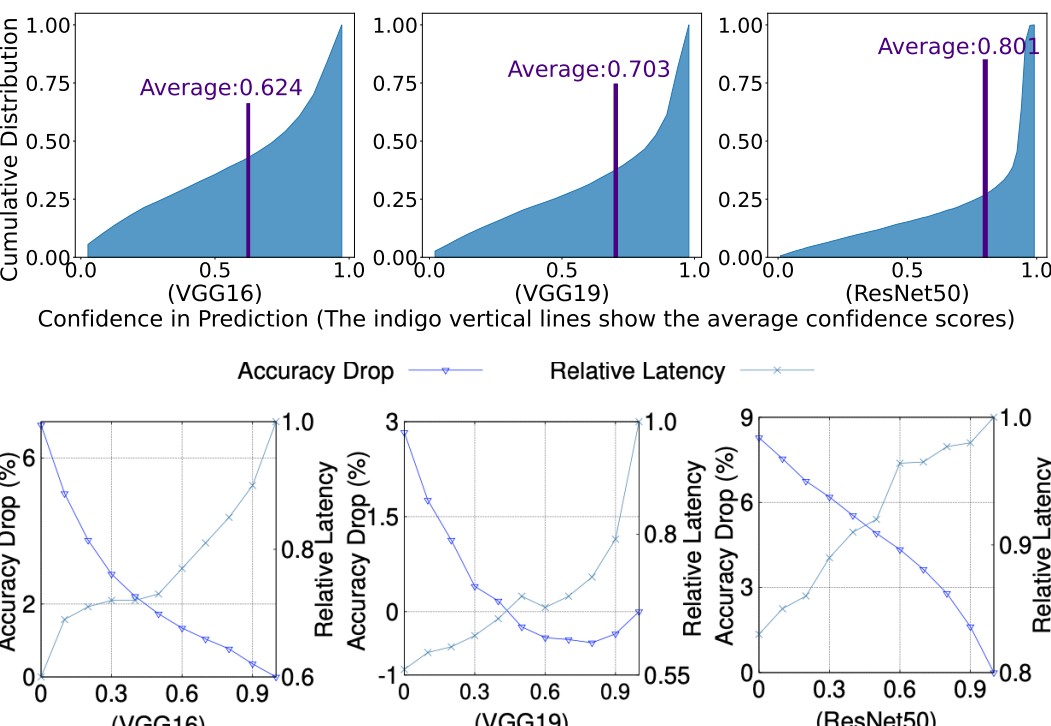

Figure 4: (Top) Cumulative distribution of the confidence values of SINF for the VGG16, VGG19, and ResNet50 DNNs. (Bottom) Accuracy, vs relative inference time and confidence threshold.

best approach *Sensitivity*. On average, SINF achieves 3.65%, 4.25%, and 2.36% better accuracy than the second-best approaches for VGG16, VGG19, and ResNet50 respectively.

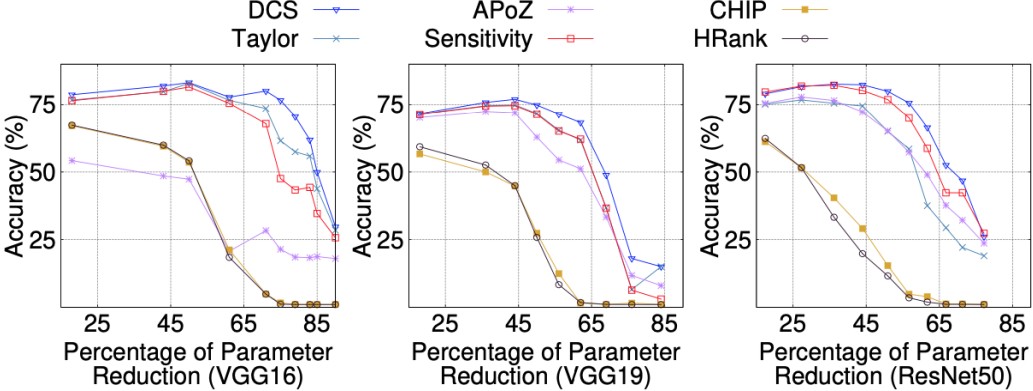

Figure 5: Comparison between DCS and state of the art for VGG19, VGG16, and ResNet50.

**DCS as Pruning Criterion.** Viewing the dataset $\mathcal{D}$ as a single macro-cluster DCS can be applied to determine the most relevant filters effectively acting as a pruning criterion. For comparison, we consider the state-of-the-art *IterTVSPrune* by Murti et al. (2023) published at ICLR 2023, which also does not require fine-tuning. For a fair comparison, we have taken the percentage of parameters pruned by *IterTVSPrune* at each layer and set the same pruning threshold for DCS. Table 1 summarizes the performance achieved by *DCS* and *IterTVSPrune*. We did not compare performance on CIFAR100 with ResNet50 as the authors of Murti et al. (2023) did not provide the performance of their approach on ResNet50 trained with CIFAR100. We notice that for different DNN structures and datasets, DCS achieves substantially better performance in 3 out of 5 settings considered while achieving similar performance in the remaining 2 settings. For VGG19, both our technique and the IterTVSPrune have pruned a significant amount of weights – respectively about 50% and 60% for CIFAR10 and CIFAR100 – possibly causing the DNN to reach a lower bound on its predictive capability thereby causing similar performance of both techniques.

Table 1: Using DCS as a Pruning Criterion vs *IterTVSPrune* (ICLR 2023).

| DNN | Dataset | Pruned | Criterion | Accuracy Loss | Difference |
|------|---------|--------|-----------|---------------|------------|
| VGG16 | CIFAR100 | 40.2% | IterTVSPrune | 18.59% | +9.75% Accuracy |
| | | 43.8% | DCS | 8.84% | -3.6% Parameters |
| VGG16 | CIFAR10 | 37.6% | IterTVSPrune | 1.9% | +0.61% Accuracy |
| | | 42% | DCS | 1.29% | -4.4% Parameters |
| VGG19 | CIFAR100 | 59% | IterTVSPrune | 5.2% | +0.05% Accuracy |
| | | 59% | DCS | 5.15% | +0% Parameters |
| VGG19 | CIFAR10 | 49% | IterTVSPrune | 1.3% | +0.4% Accuracy |
| | | 49.65% | DCS | 0.9% | -0.65% Parameters |
| ResNet50 | CIFAR10 | 34.1% | IterTVSPrune | 9.94% | +8.13% Accuracy |
| | | 39.92% | DCS | 1.81% | -5.82% Parameters |
| ResNet50 | ImageNet | 9.98% | IterTVSPrune | 10.21% | +2.41% Accuracy |
| | | 12% | DCS | 7.8% | -2.02% Parameters |

**Per-Cluster Accuracy Gain.** We posed ourselves the following question: "*Can SINF perform better than the original DNN when considering the accuracy obtained in individual clusters?*". We find the subgraphs the with lowest percentage of parameters retained while satisfying the constraint on the evaluation criterion (here it is accuracy) posed in 1. Figure 6 shows the accuracy gain obtained on the individual clusters by those subgraphs as compared to the original VGG16, VGG19, and ResNet50 DNNs. Intriguingly, *SINF provides on the average 5.73%, 8.38% and 6.36% better per-cluster accuracy than the original VGG16, VGG19, and ResNet50 DNNs, respectively, notwithstanding that the number of parameters have been reduced by 30%, 50%, and 44%*. We believe the reason behind this improvement is that the semantic partitioning performed by SINF improves the DNN explainability saving the DNN from being "less confused" among different semantic clusters, which justifies better results when considering per-cluster accuracy.

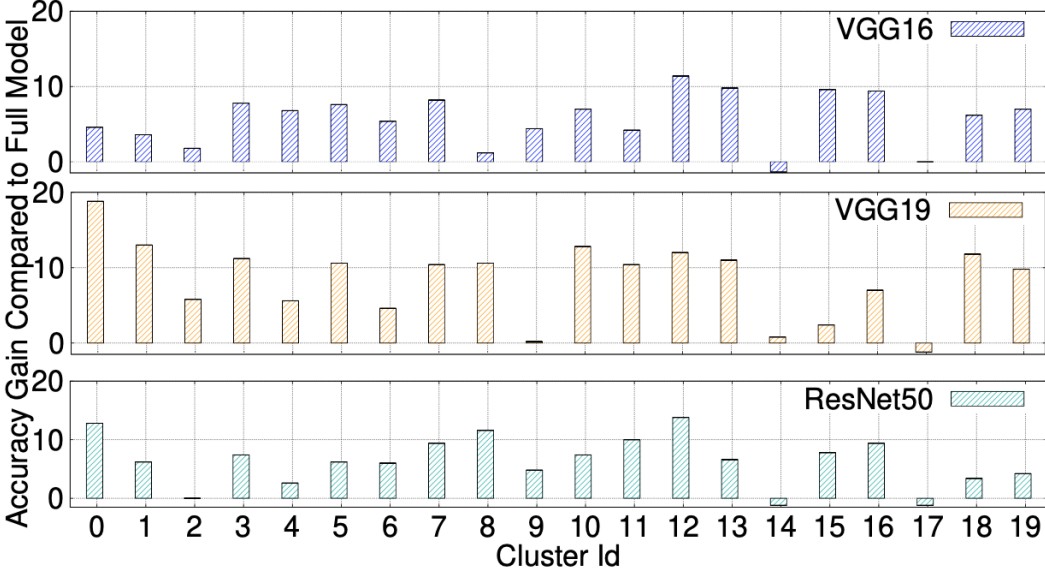

Figure 6: Performance gain compared to the original DNN. The x-axis shows the ids of different semantic clusters from CIFAR100 dataset and y-axis shows the performance improvement when models corresponding to specific classes are used.

## 7 CONCLUDING REMARKS

In this paper, we have proposed a new framework called *Semantic Inference* (SINF) which allows to achieve faster execution of DNNs without compromising accuracy. As part of SINF, we have proposed a new approach named *Discriminative Capability Score* (DCS) to find subgraphs inside large DNNs to discriminate among the members of a specific semantic cluster. We have benchmarked the performance of SINF on the VGG16, VGG19 and ResNet50 DNNs trained on the CIFAR100 and CIFAR10 datasets. By comparing the performance of SINF with several existing approaches, we have shown that SINF outperforms the state of the art.

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

## A  APPENDIX

We provide further result for SINF on ImageNet in section B. We list all the notations in section C for ease of reference.

## B  EXPERIMENT ON IMAGENET

In this section we provide the performance comparison of DCS with the *"Taylor"* (Molchanov et al., 2019) as this proved to be the second best method in our experiment with CIFAR100. We choose a subset of ImageNet and create six semantic clusters namely - Bird, Lizard, Animal, Insect, Seafish, salt-water carnivores. We plan to delve into the task of defining semantic similarity among the classes of ImageNet and providing more extensive results if the work is accepted or in extension of this work. Table 2 provides the list of classes in each cluster of ImageNet that we are considering here. From figure 8 we can see that DCS does significantly better than Molchanov et al. (2019). On an average, DCS does 3.85% better than *"Taylor"* metric. Figure 7 shows the change in accuracy and relative latency with different confidence thresholds. On an average, we can achieve 20%, and 27% lower latency on the considered subset of ImageNet with VGG16 and VGG19 respectively.

## C  TABLE OF NOTATIONS

Table 3 list all the notations used in this work. We provide a general introduction to the notations introduced in different sections here. Detailed description of the notations can be found in their respective sections.

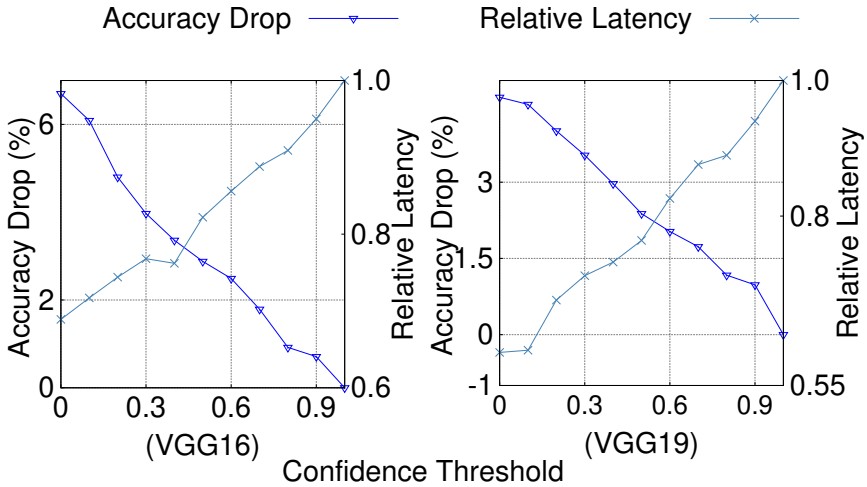

Figure 7: Comparison of DCS and Taylor on Subset of ImageNet dataset.

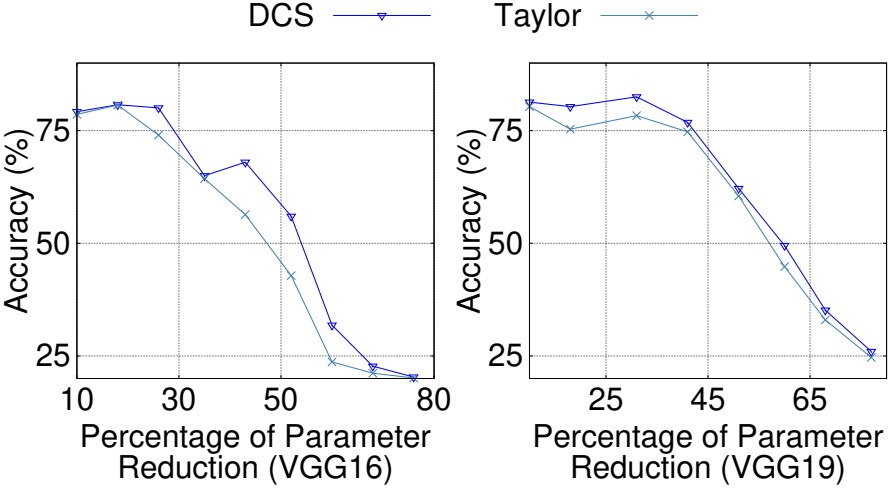

Figure 8: Comparison of DCS and Taylor on Subset of ImageNet dataset.

Table 2: Subset of ImageNet for experiment with DCS

| Semantic Cluster Name | Member Fine Classes |
|---|---|
| Bird | brambling, Fringilla montifringilla
goldfinch, Carduelis carduelis
house finch, linnet, Carpodacus mexicanus
junco, snowbird
indigo bunting, indigo finch, indigo bird, Passerina cyanea |
| Lizards | box turtle, box tortoise
banded gecko
common iguana, iguana, Iguana iguana
American chameleon, anole, Anolis carolinensis
whiptail, whiptail lizard |
| Animals | basenji
keeshond
hyena, hyaena
Persian cat
mongoose |
| Insects | dung beetle
fly
bee
ant, emmet, pismire
grasshopper, hopper |
| Sea-fish | anemone fish
sturgeon
gar, garfish, garpike, billfish, Lepisosteus osseus
lionfish
puffer, pufferfish, blowfish, globefish |
| salt-water carnivores | great white shark, white shark, man-eater, Carcharodon caharias',
tiger shark, Galeocerdo cuvieri
hammerhead, hammerhead shark
electric ray, crampfish, numbfish, torpedo
stingray |

Table 3: Notations Used in this Work

| Symbol | Identification | Symbol | Identification |
|---|---|---|---|
| $\mathcal{B}_{eval}$ | Evaluation criterion of the sub-graphs | $\mathcal{D}$ | Complete Dataset |
| $\gamma_i$ | $i$-th semantic cluster | $\mathcal{D}_{\gamma_i}$ | Dataset corresponding to semantic cluster $\gamma_i$ |
| $\mathcal{F}$ | Base DNN | $\mathcal{F}_{\gamma_i}$ | Sub-graph corresponding to semantic cluster $\gamma_i$ |
| $X^j$ | $j$-th sample in the dataset | $t^j$ | ground truth label corresponding to the $j$-th sample |
| $\mathbf{A}_{l,c_i}^j$ | Activation map of $l$-th layer's $c_i$-th filter for $j$-th sample in the dataset | H,W | Respectively height and width of an activation map |
| $C_{out}^l$ | Number of output channels in layer $l$ of a DNN | $\mathcal{P}(\cdot)$ | Adaptive Pooling Operation |
| $\mathbf{F}_{l,c_i}^j$ | Feature vector obtained for $l$-th layer's $c_i$-th filter and $j$-th sample | $\mathbf{F}_l^j$ | Feature vector of layer $l$ (After Concatenating $\mathbf{F}_{l,c_i}^j$ for all filters) |
| $N_f$ | Length of a feature vector | $N_c$ | Number of classes |
| $\mathbf{W}$ | Transformation Matrix | $\mathbf{W}_1^*$ | Optimized transformation matrix for layer $l$ |
| $\mathcal{L}_{DOF}$ | Objective Function | $\mathbf{I}^l$ | Feature importance matrix of layer $l$ |
| $s_i$ | Norm of the $i$-th column of $\mathbf{I}^l$ (Importance of $i$-th feature) | $\mathbf{s}_l$ | Feature importance vector of layer $l$ |
| $\mathbf{u}_{c_i}$ | Feature importance vector corresponding to $c_i$-th filter of a layer | $\mathcal{G}$ | Group norm operation |

