# OpenReview forum: "Faster and Accurate Neural Networks with Semantic Inference"
_ICLR.cc/2024/Conference — Submitted to ICLR 2024_

### Official Review · Reviewer_wsaJ · 2023-10-21

**Soundness:** 2 fair
**Presentation:** 1 poor
**Contribution:** 3 good
**Rating:** 5
**Confidence:** 2

**Summary:**

The authors propose a method to make DDN inference more efficient by selecting for each input, using the features of early layers, which units to use in the rest of layers.
They do so by finding clusters in the training dataset such that each cluster is assigned a subnetwork within the whole DNN.
At inference time, a sample is assigned to a cluster and, thus, to a subnetwork.
The result show that a substantial reduction in inference time can be obtained while sacrificing some accuracy on CIFAR100 using VGG and ResNet DNNs.

**Strengths:**

With the activations in the high-level layers of DNNs (and not so much the lower-level ones) known to be sparse, it seems reasonable to perform predictive pruning conditioned on the low-level activation of a sample.

**Weaknesses:**

Although I understand the motivation and the gist of the method (Fig 3 conveys it quite well), I struggled to follow some of the details. These are my main issues with the paper:
1- In Eq 1, is L_eval is a loss (where lower is better), it seems trivial to find subnetworks that perform less well than the original DNN. So I imagine L_eval is not a loss (as the L would suggest), but some performance metric where higher is better. Even in that case the problem is not fully defined, since F_yi = F would satisfy Eq 1 while not being an useful solution.
2- I find the writing of the paper very hard to follow. How are the semantic clusters built? I’m not sure I could follow the explanation in section 4 and algo 1, which is a bit too cluttered. Something as important for the method as the Discriminative Capability Score is not actually explained anywhere. Same issue with the training of the route predictor.
3- Even in section 3, which introduces the main setting of the methodology, the authors already mention many elements that would rather pertain to the experimental section, like specific architectures and hyperparameter choices.
4- The authors claim to obtain SOTA results while using relatively old datasets and architectures. I don’t think such a choice invalidates the contribution, but I would weigh the claims accordingly.
5- What is Fig 1top supposed to convey? It doesn’t seem to show anything. If the idea is to show the difference in sparsity, a simply plot with the sparsity level of each layer would work better.

**Questions:**

Overall, I think it is likely that the contribution of this paper could be valuable, but the presentation and understandability needs to be substantially improved.

The authors would probably be interested in this paper:
Ye, Mao, Chengyue Gong, Lizhen Nie, Denny Zhou, Adam Klivans, and Qiang Liu. "Good subnetworks provably exist: Pruning via greedy forward selection." In International Conference on Machine Learning, pp. 10820-10830. PMLR, 2020.

---

> ### Author Response · Authors · 2023-11-19
>
> > In Eq 1, is L_eval is a loss (where lower is better), it seems trivial to find sub-networks that perform less well than the original DNN. So I imagine L_eval is not a loss (as the L would suggest), but some performance metric where higher is better. Even in that case the problem is not fully defined, since F_i = F would satisfy Eq 1 while not being a useful solution.
>
>  We thank the reviewer for this valuable comment. Yes, the evaluation metric shown in Semantic DNN Subgraph Problem (SDSP) does not correspond to the loss function. We define it as the evaluation criterion used for scoring the network and its subgraphs on their respective datasets. In our work, we have chosen the evaluation metric as accuracy. We understand the confusion regarding the notation which would not arise if the property of the evaluation metric (higher value means better performance) was mentioned. We have updated the manuscript accordingly. We have changed the notation of the evaluation criterion to $\mathcal{B}_{eval}$ and added
>
> *A higher value of $\mathcal{B}_{eval}$ is assumed to correspond to better performance*.
>
> As mentioned in the comment, $\mathcal{F}_{\gamma_i} = \mathcal{F}$ would satisfy Equation 1. In our case, the problem definition would be more precise if we used the word *proper subgraph*. This would exclude the trivial solution and would solve the confusion. We have modified the problem statement to incorporate this discussion.
> > I find the writing of the paper very hard to follow. How are the semantic clusters built? I’m not sure I could follow the explanation in section 4 and algo 1, which is a bit too cluttered.
>
> We are sorry that Section 4 and Algorithm 1 have been hard to follow. We have updated the manuscript and tried to make the steps easier to follow. We hope that it will clarify the questions that the reviewer might have regarding the procedure.
> > Something as important for the method as the Discriminative Capability Score is not actually explained anywhere.
>
> Section 4 describes the procedure to obtain the discriminative capability score. The section builds up the intuition behind the discriminative capability score and describes it right before the equation 3. We have updated the manuscript to make it easy to follow the procedure and find the description.
> > Same issue with the training of the route predictor**
>
> We have added more details to the *Semantic Route Predictor* subsection of Section 5 to illustrate the training of the semantic route predictor. Specifically, we add that
>
> *To train the auxiliary classifier $\boldsymbol{\chi}$, the section of the base model up to the $M-1$-th layer of $\mathcal{F}$ is frozen and the classifier is trained in supervised fashion using {$\mathbf{A}_{M-1}^j, \gamma_m^j$} , where  j=1...|D| as the dataset.*
>
> *Here, $\mathbf{A}_{M-1}^j$, and $\gamma_m^j$ are respectively the activation of the $M-1$-th layer of the base model and the ground truth semantic cluster for the j-th sample. As we are considering a pre-trained base DNN, we train the auxiliary classifier separately from the base network using the activations obtained from the $M-1$-th layer.*
>
> > The authors claim to obtain SOTA results while using relatively old datasets and architectures. I don’t think such a choice invalidates the contribution, but I would weigh the claims accordingly.
>
> We thank you for your comment. The setting we are proposing is new and it requires a score-based approach to select the filters in each layer to form the subgraph. As a result, we have chosen the [1], [2] for comparison as recent methods. Apart from that, we compare with the recent [3] which is a published work in ICLR 2023 to prove the efficacy of DCS as a pruning method. We are currently running SINF on ImageNet-1k and will report the results as soon as we obtain them.
>
> [1] Sui, Y., Yin, M., Xie, Y., Phan, H., Aliari Zonouz, S. and Yuan, B., 2021. Chip: Channel independence-based pruning for compact neural networks. Advances in Neural Information Processing Systems, 34, pp.24604-24616.
> [2] Lin, M., Ji, R., Wang, Y., Zhang, Y., Zhang, B., Tian, Y. and Shao, L., 2020. Hrank: Filter pruning using high-rank feature map. In Proceedings of the IEEE/CVF conference on computer vision and pattern recognition (pp. 1529-1538).
> [3] Murti, C., Narshana, T. and Bhattacharyya, C., 2022, September. TVSPrune-Pruning Non-discriminative filters via Total Variation separability of intermediate representations without fine tuning. In The Eleventh International Conference on Learning Representations.

---

> > ### Author Response · Authors · 2023-11-19
> >
> > > What is Fig 1 top supposed to convey? It doesn’t seem to show anything. If the idea is to show the difference in sparsity, a simply plot with the sparsity level of each layer would work better.
> >
> > Figure 1 (top) shows the filter activations for two layers of ResNet50. The figure aims at clearly conveying the sparsity level of each layer. We also focus attention on the similarity of the two classes (otter and seal) in the earlier layer and the distinctness in the later layer. We also quantify this based on the L1 distance. In the earlier layer, the $L_1$ distance is 0.028 while the same for the later layer is 0.111. This shows the specialization of filter activation in deeper layers. From this, we conclude that the deeper layers are more sparse and more specialized for individual classes.

---

> ### Comment · Reviewer_wsaJ · 2023-11-22
>
> I thank the authors for their responses. I would like to clarify and follow up on some of the issues.
>
> - Eq (1): I'm not sure writing "a proper subgraph" solves the issue. If I understand well an efficiency is the main objective, it would make more sense to aim at minimizing the number of nodes in each $\mathcal{F}_{\gamma_i}$ subject to Eq (1). I would suggest to revise Section 3 accordingly.
> - Clarity: I apologize if my comment was unclear. I meant that DCS is not introduced as a high-level concept, and its connection to solving Eq (1) is not explained in an intuitive manner, with Algo. 1 being so cluttered it makes it very hard for a reader to understand the gist. I would strongly suggest the authors to revise the whole paper to allow for a better flow, such that every section/paragraph leads naturally to the following. Similarly, Fig 1 needs to be improved as per my comment in the previous round.

---

### Official Review · Reviewer_n596 · 2023-10-23

**Soundness:** 2 fair
**Presentation:** 1 poor
**Contribution:** 2 fair
**Rating:** 3
**Confidence:** 4

**Summary:**

This work introduces Semantic Inference (SINF), a novel method for drastically reducing the inference complexity of DNNs. SINF is based on the assumption that semantically similar inputs should share a significant number of filter activations; these semantically similar inputs can be viewed as "clusters". In turn, appropriate cluster-specific sub-graphs of the base network can be detected and then executed for inference based on the assigned cluster of each object. To do so, the considered approach introduces Discriminative Capability Score (DCS)  , a general purpose method aiming to find filters that can distinguish between semantically similar classes. In experimental evaluations on the CIFAR-10/100 datasets using different benchmark architectures (VGG16, VGG19, RN50), DCS yielded significant improvements compared to other SOTA methods.

**Strengths:**

The paper introduces a novel approach for using sub-graphs of the network to perform inference based on a partition of the inputs via semantic similarity. The idea is very interesting and intuitive, potentially contributing to a different inference methods for DNNs.

**Weaknesses:**

Despite the novelty of the approach, there are some significant issues that need to be addressed, and the main element missing from the main text, at least in my personal view, is clarity. Up until (and including) Section 3, the paper is well-written and easy to follow. The intuition and motivation are very clear.

After that, the presentation of the method is very confusing.

We begin with the definition of the DCS procedure, where the notation keeps changing. The authors start with a given layer $l$ and semantic cluster $\gamma_m$ (where up to this point, it is not clear if $\gamma_m$ comes from some ground truth information or is computed as the introduction suggested). For each datapoint belonging to the semantic cluster, the activations for each feature map $\boldsymbol A_{l, c_i}^j$ are computed and adaptive pooling is perfomed. Then, the feature map is flattened by the dependence on $l$ is gone. This is the first instance that breaks the flow of reading the paper. This trends continues by introducing $\boldsymbol F^j$ which concatenates all the features for all the filters in the layer and $N_f$ is introduced (again layer-specific with no dependence). The same goes for $\boldsymbol W*$ that seems to be dependent on both $\gamma_m$ and $l$. Using some normalization, we then compute the desired DCS value, which again is layer and cluster specific but this is not clear from the notation.

Then, the SINF procedure is presented. The authors present the steps, most of which are not yet introduced. It would be better to first introduce the procedures and then tie them together, instead of noting that everything will be explained later.

Here, the authors note that before deployment, the DCS is used to construct semantic subgraphs. Again is not yet clear how are the partitions obtained. Do the authors use the training set of CIFAR-10/100 that are already somehow prepartitioned into semantic clusters, e.g., the superclasses of CIFAR-100, and use algorithm 2 to extract the subgraphs? Because this is the first step of the approach, that always takes as an input the Partitioned Data.

Moving on to the process of extracting the sub-graphs, we begin with $L$ and $M$, the last layer and the layer before the Common Feature Extractor. In this context, if $L$ is the last layer of (assuming) the backbone network, isn't $M$ also the layer of said network? In Fig.3 there is no illustration of what $L$ and $M$ since they backbone is not shown. The input is just passed through the common feature extractor and a sub-graph is selected without a corresponding backbone.

The authors use $X$ to denote the number of retained filters in a layer $X$. Again confusing notation, $X$ are the data, and even though it's easy to understand the difference, clarity is reduced. The authors can just use $r_l$ for any layer $l$. The authors note that for each layer $M \leq l \leq L$ the $r_l$ is calculated; however, in Algorithm 2, $r_L$ and $r_M$ are given as an input and also updated in the process. At the same time the authors note "This procedure is performed for different values of $r_L$ and $r_M$. In this paper, $r_L$ is set between 90% and 10%, with steps of 10, while $r_M$ is set between 10% and 1%, with steps of 2". Why is this procedure performed? Is this some kind of initialization? What happens with the multiple different values? Do the authors select some kind of best performing model? And in this context, what is the intuition behind the computation of $r_l$? It seems to be dependent on the current layer number $M, L, r_M$, and  $r_L$ but how the values affect the results is not clear.

Turning to the classifier, the authors note that the classifier is attached to some $l$-th layer. This layer is the earliest layer that provides good prediction for semantic routes is chosen. What does it mean that the layer provides good predictions for semantic routes? Do the authors try to match the output of this classifier to the potential $K$ clusters (which at this point I will assume stem from ground truth information)? How was the $75$% accuracy was decided? What if for some setting the $l$-th layer is the last layer of the network? What is the impact of different values for this threshold?

In the experimental section, the authors introduce the notation for the confidence threshold as $\gamma$, which is also used for the partitions $\gamma_i$. Again, even though it is something distinct, notation clarity should be improved.

Other points:

In the context of pruning methods, there exist a plethora of methods that perform pruning end-to-end during training, balancing accuracy and sparsity. Characteristic examples include [1,2,3]. However, one argument that the authors make is that all the (referenced) methods require fine-tuning after pruning, which may be true for the methods that the authors cite but not for all pruning methods. Unless the authors refer to post-hoc pruning methods, in which again there exist methods that consider solvers to balance accuracy and sparsity without any further fine-tuning [4].

I find the definition of the Semantic DNN Subgraph Problem a bit misdefined. The equation itself seems to indicate that the average loss of the partitions should be less or equal to the full network loss; this however can be true for any configuration of the sub-graphs (even random) and restricts solutions that can even perform better, e.g., due to avoiding parameter redundancy. The authors note that $\mathcal{L}$ is the evaluation metric, but the $\mathcal{L}$ notation is commonly the loss function in the literature and not the classification accuracy that the authors consider. And in the regression setting, e.g., using MSE as an evaluation metric leads to a different interpretation of the problem setting.

The fact that the evaluation is only assessed on CIFAR-10/100 undermines the impact of the approach. Indeed, these datasets are considered "small" datasets and the authors should assert the generalization capabilities and performance on more demanding datasets and settings, such as ImageNet-1k. At the same time, evidently the algorithms depends on ground truth information about the semantic clusters of the data. If we do not have access to this kind of information, it's not clear: (i) what semantic clusters we should consider, and (ii) how will the algorithm perform in this setting.

Overall, despite the novelty of the proposed framework, presentation and clarity should be greatly improved. At the same time, important  details (like the $r_l$ computation) are not justified, while important ablation studies in the various effects of parameters/settings are missing.

[1] Louizos et al., Bayesian Compression for Deep Learning, In Proc. NIPS 2017

[2] Panousis et al., Nonparametric Bayesian Deep Networks with Local Competition, In Proc. ICML 2019

[3] Neklyudov et al., Structured bayesian pruning via log-normal multiplicative noise, In Proc. NIPS 2017

[4] Wong et al., Leveraging sparse linear layers for debuggable deep networks. In Proc. ICML, 2021.

**Questions:**

Please see the Weaknesses section.

---

> ### Author Response · Authors · 2023-11-19
>
> > The authors start with a given layer $l$ and semantic cluster $\gamma_m$ (where up to this point, it is not clear if $\gamma_m$ comes from some ground truth information or is computed as the introduction suggested [...] the authors note that before deployment, the DCS is used to construct semantic subgraphs. Again is not yet clear how are the partitions obtained.
>
> Thank you for your comment. In Section 3, we have stated that we assume that the $\gamma_m$ clusters are defined based on application-level similarities (e.g., classes related to flowers, insects, etc.) or pre-defined at the dataset level (e.g., as in the CIFAR100 dataset). In this case, the 20 superclasses in CIFAR100 form the semantic clusters we are considering. We are using this dataset to experiment on especially because it is pre-partitioned into semantic clusters. We have remarked this aspect in the updated manuscript we have submitted as part of the rebuttal.
>
> > For each data-point belonging to the semantic cluster, the activations for each feature map $A_{l,c_i}^j$ are computed and adaptive pooling is performed. Then, the feature map is flattened by the dependence on $l$ is gone.
>
> We are sorry to see that the notation might have been confusing. Algorithm 1 is designed for a specific cluster and a specific layer and it is specifically mentioned in the algorithm heading. That is why the dependence on the layer and cluster is not explicitly mentioned for the feature vector and subsequent symbols. The dependence is maintained in the notation for the other quantities before so as not to disrupt the already defined quantities. However, we agree that keeping the dependence on the layer and semantic cluster can make it easier to follow the algorithm. We have made changes to reflect the dependence on layer and semantic cluster in the manuscript and we have reuploaded it for the reviewer’s convenience.
> > Moving on to the process of extracting the sub-graphs, we begin with $L$ and $M$, the last layer and the layer before the Common Feature Extractor. In this context, if $L$ is the last layer of (assuming) the backbone network, isn't $M$ also the layer of said network? In Fig.3 there is no illustration of what $L$ and $M$ since they backbone is not shown. The input is just passed through the common feature extractor and a sub-graph is selected without a corresponding backbone.
>
> The $L$ and $M$ are part of the same backbone network. As defined in Section 3, we extract the sub-graph from a pre-trained network. For the sake of adaptive inference, we keep the first part as a common feature extractor and the rest of the network is used for the specialization for different semantic clusters. In this case, $L$ is the last layer of the backbone network and $M$ is the layer just before the common feature extractor from the end.
>
> The first $l$ layers of the backbone network are frozen and used as the common feature extractor. The motivation is that the earlier layers of a DNN focus on features which overlap among multiple classes and possibly semantic clusters. From that perspective, $M$ would be the $l+1\ th$ layer of the back-bone and $L$ would be the last layer as mentioned. Through experimentation, we have found that at least the first 5 convolution layers are required to obtain a reasonable prediction accuracy on the semantic clusters for VGG16 and VGG19 and the first 2 layers are required for ResNet50. Therefore, the value of $M$ is 6 for VGG models and 22 for ResNet50. We have updated the manuscript clarifying the relation between the base DNN, the common feature extractor and the extracted sub-graph.
>
> > The authors note that "This procedure is performed for different values of $r_L$ and $r_M$. In this paper, $r_L$ is set between 90\% and 10\%, with steps of 10, while $r_M$ is set between 10\% and 1\%, with steps of 2". Why is this procedure performed? Is this some kind of initialization? What happens with the multiple different values? Do the authors select some kind of best performing model?
>
> We thank the reviewer for pointing this out and we have fixed the notation in the updated version of the manuscript we have uploaded as part of the rebuttal.
>
> The procedure of extracting sub-graph is performed for varying values of $r_L$ and $r_M$. This is because there can be multiple sub-graphs that satisfy our accuracy requirement. However, those sub-graphs will vary in their size and latency. By trying  different values of $r_L$ and $r_M$, we can obtain several solutions satisfying our performance requirement and choose the optimum one based on additional requirements, for example, sub-graph size, latency. In other words,  we are selecting the best performing DNN.

---

> > ### Author Response · Authors · 2023-11-19
> >
> > > What is the intuition behind the computation of $r_l$? It seems to be dependent on the current layer number $M, L, r_l$, and $l$ but how the values affect the results is not clear.
> >
> > The effect of the percentage of retained filters is directly related to the parameters retained in each layer. It is intuitive that the smaller the number of parameters, the worse the DNN might perform. Thus, we perform a grid search on the number of retained filters in each layer to find the combinations that satisfy our constraint as defined in Equation 1. Notice that changing the percentage of filters in each layer would be very costly and time consuming. Based on the observation (discussed in Section 3) that the deeper the layer, the more sparse and more specialized the activations become for the individual classes, we restrict ourselves to linearly decreasing the percentage of retained filters from the $M\ th$ layer to the $L\ th$ layer. To calculate the percentage of filter in each layer $M \leq l \leq L$ given the $M, L, r_L$ and $r_M$, we use the linear equation as shown in Algorithm 2. We have clarified this point in the updated manuscript.
> >
> > > What does it mean that the layer provides good predictions for semantic routes? Do the authors try to match the output of this classifier to the potential $K$ clusters (which at this point I will assume stem from ground truth information)? How was the 75\% accuracy was decided?
> > We are sorry for the confusion. The 75% accuracy threshold for choosing the layer $l$ is application-level constraints and in the paper was chosen as an example. We have clarified this in the current version.
> >
> > > What if for some setting the $l$-th layer is the last layer of the network? What is the impact of different values for this threshold? **
> >
> > Existing work has unveiled that for the DNN models considered, fairly good accuracy can be obtained even at early layers [1]. As a result, it is highly unlikely that the $l$-th layer will have to be chosen as the last layer. However, in this case, the concern about DNN accuracy can be averted by using the confidence threshold. If the semantic route predictor is not confident enough, we can bypass the extracted sub-graph and use the base DNN. This decision is made by the feature router. The feature router is presented with the prediction of the semantic route predictor as well as its confidence in its prediction. Provided that the confidence is smaller than the predefined threshold, the feature router activates the full DNN model. This way, we can balance the accuracy and latency trade-off. We have clarified this in the current version.
> >
> > [1] Han, D.J., Park, J., Ham, S., Lee, N. and Moon, J., 2023. Improving Low-Latency Predictions in Multi-Exit Neural Networks via Block-Dependent Losses. IEEE Transactions on Neural Networks and Learning Systems.
> >
> > > In the experimental section, the authors introduce the notation for the confidence threshold as $\gamma$, which is also used for the partitions $\gamma_i$. Again, even though it is something distinct, notation clarity should be improved.
> >
> > We thank the reviewer for pointing this out. We have updated the manuscript and introduced different symbols to differentiate between the confidence threshold and the semantic cluster to improve clarity.
> > > Unless the authors refer to post-hoc pruning methods, in which again there exist methods that consider solvers to balance accuracy and sparsity without any further fine-tuning [4].
> >
> > Yes, we are considering the post-hoc pruning only. To be exact, as mentioned in Section 3 (Semantic DNN Subgraph Problem), we are considering the $\mathcal{F}$ to be trained on a dataset $\mathcal{D}$. After that, we are not changing the weights of the DNN further. Instead, we are only extracting the sub-graphs for the semantic clusters. In stark opposition, the paper [4] only focuses on the sparsity of the decision layer for explainability, which is out of the scope of our work.
> > > I find the definition of the Semantic DNN Subgraph Problem a bit misdefined. The equation itself seems to indicate that the average loss of the partitions should be less or equal to the full network loss; this however can be true for any configuration of the sub-graphs (even random) and restricts solutions that can even perform better, e.g., due to avoiding parameter redundancy.
> >
> > We thank you for pointing this out. The evaluation metric shown in Semantic DNN Subgraph Problem (SDSP) does not directly correspond to the loss function. Conversely, we define it as the evaluation criterion used for scoring the DNN and its subgraphs on their respective datasets. In our work, we have chosen the evaluation metric as accuracy. We show in Section 6 (Per-cluster Accuracy Gain) that our extracted sub-graphs can indeed perform better than the original DNN on the respective semantic clusters.

---

> > > ### Author Response · Authors · 2023-11-19
> > >
> > > > The authors note that $\mathcal{L}$ is the evaluation metric, but the $\mathcal{L}$ notation is commonly the loss function in the literature and not the classification accuracy that the authors consider. And in the regression setting, e.g., using MSE as an evaluation metric leads to a different interpretation of the problem setting.
> > >
> > > We understand the confusion that can arise due to the notation used to denote the evaluation criterion. We thank the reviewer for pointing that out. We have updated the manuscript to represent the evaluation criterion using $\mathcal{B}_{eval}$ to improve clarity.
> > >
> > > We also add the assumption that a higher value of  $\mathcal{B}_{eval}$  corresponds to better performance. We emphasize that this assumption does not hinder in any way the use of MSE or a similar evaluation criterion, as one can take the negative of the value obtained and use the same constraint described in the problem statement. We have added these explanations in the manuscript.
> > > > The fact that the evaluation is only assessed on CIFAR-10/100 undermines the impact of the approach. Indeed, these datasets are considered "small" datasets and the authors should assert the generalization capabilities and performance on more demanding datasets and settings, such as ImageNet-1k.
> > >
> > > We are currently running SINF on ImageNet-1k and will report the results as soon as we obtain them.

---

### Official Review · Reviewer_NR9c · 2023-11-01

**Soundness:** 2 fair
**Presentation:** 3 good
**Contribution:** 3 good
**Rating:** 6
**Confidence:** 3

**Summary:**

1) Key ideas: the submitted paper studies the problem of pruning approaches of deep neural networks, with a focus on searching cluster-specific subgraphs for inference (without retraining or fine-tuning) that faster neural network without drastic accuracy loss.

2) Contributions: The authors propose a new inference framework to divide the DNN into subgraphs according to the semantic cluster the object belongs to. This process is complemented by common feature extractor, semantic route predictor and feature router modules.  Additionally, a new discriminative capability score (DCS) is proposed to find the subgraphs, which is also applied as a pruning criterion and achieve state-of-the-art performance.

3) Their significance: the most significant contribution is the results: 1. DCS outperforms existing state-of-the-art discriminative algorithms and pruning criterion. 2. SINF reduces the inference time with limited accuracy loss and showing significant improvement considering per-cluster accuracy.

**Strengths:**

1. The paper shows impressive results of reducing inference time with limited accuracy loss (Figure 4 and Figure 6) and the proposed DCS shows state-of-the-art performance (Figure 5 and Table 1).
2. Visualization and quantization analysis and verification have been conducted on the proposed Semantic DNN Subgraph Problem. Figure 1 shows intuitive and credible results.
3. The paper provides detailed descriptions of the proposed core algorithms (how to extract subgraphs for semantic clusters, algorithm2) and theories (how to compute discriminative capability score, algorithm1).

**Weaknesses:**

1. Lack the comparison with previous quantization and pruning approaches. There are many quantization and pruning approaches that can speed up the inference without fine-tuning or retraining. Can do more comparison experiments with SINF and show the comparable results of the whole pipeline.
2. Table 1 lack the results of ResNet50 on CIFAR100.
3. Lack the analysis of the poor improvement in Table1, where there is almost no improvement for VGG19 on CIFAR100 and only a slight improvement on CIFAR10.

**Questions:**

Is there any proof or metric to evaluate the intrinsic redundancy in latent representations? The idea of intrinsic redundancy is interesting but somehow blurred, it will be great if there is any explicit metric can be proposed or discussed.

---

> ### Author Response · Authors · 2023-11-19
>
> # Regarding Weakness 1
> > There are many quantization and pruning approaches that can speed up the inference without fine-tuning or retraining. Can do more comparison experiments with SINF and show the comparable results of the whole pipeline.
>
> We thank you for your comment. To the best of our knowledge, the previous work that does not require fine tuning is very limited -- we were only able to find [1] as a recent paper that does not require fine-tuning. We would be happy to compare against other non fine-tuning techniques if the reviewer shared these papers with us. We would like to clarify that *we are not considering pruning at train-time or retraining approaches*. We are assuming that we start with a pre-trained DNN and extract the sub-graphs corresponding to the semantic clusters defined by the task at hand.
>
> For the extraction of the sub-graph part, we have compared against metrics which are usable in our scenario. For the pruning part, we are considering the most recent work closest to our setting for comparison. Notwithstanding, we point out that designing a pruning algorithm is out of the scope of this paper. Our goal is to design a dynamic DNN which can *immediately* adapt to changing requirements. For this reason, we cannot fine-tune the DNN at run time.
>
> Regarding quantization, as pointed out in the background section (Section II), we believe our proposed work is orthogonal to that approach and can be used on top of quantization and coding approaches to further improve the performance.
>
> [1] Murti, C., Narshana, T. and Bhattacharyya, C., 2022, September. TVSPrune-Pruning Non-discriminative filters via Total Variation separability of intermediate representations without fine tuning. In The Eleventh International Conference on Learning Representations.
>
> # Regarding Weakness 2
> > Table 1 lacks the results of ResNet50 on CIFAR100.
>
> Thank you for pointing this out. We are comparing our approach to the work which has been published in the last ICLR 2023 [1] . To make a fair comparison, we are using the same DNN model as in [1]. Since the work does not provide any performance result for ResNet50 on CIFAR100, we are not comparing for the same.
>
> # Regarding Weakness 3
> > Lack the analysis of the poor improvement in Table 1, where there is almost no improvement for VGG19 on CIFAR100 and only a slight improvement on CIFAR10.
>
> We thank you for the opportunity to clarify this point. A possible explanation is that since both our technique and the state of the art IterTVSPrune have pruned a significant amount of weights – respectively about 50% and 60% for CIFAR10 and CIFAR 100% – the DNN has reached a lower bound on its predictive capability. In other words, this means that the DNN cannot be pruned more without compromising the accuracy. In the other DNNs (VGG16 and ResNet50) the amounts of weights pruned is less aggressive (up to 40% for VGG16 and up to 35% for ResNet50) so eventually there could be room for improvement. This aspect is nevertheless intriguing and we plan to delve deeper into this aspect in future work. We will include these discussions in the final manuscript.
>
> # Regarding the Question
> > Is there any proof or metric to evaluate the intrinsic redundancy in latent representations?
>
> In Section 3, we have used $L_1$ distance to quantify the similarity of the filter response for inputs of two different classes (otter and seal). We have also utilized the overlapping between filter activation for two classes to quantify this similarity. Although these metrics do not exactly quantify the intrinsic redundancy, it is part of our current work to characterize the redundancy in a more formal manner.

---

### Official Review · Reviewer_wsBg · 2023-11-04

**Soundness:** 2 fair
**Presentation:** 2 fair
**Contribution:** 2 fair
**Rating:** 3
**Confidence:** 4

**Summary:**

The paper introduces the Semantic Inference (SINF) framework to accelerate the execution of DNN. Leveraging the intrinsic redundancy in latent representations, the authors propose the Discriminative Capability Score (DCS) to identify subgraphs within large DNNs to discriminate between members of specific semantic clusters. They validate the approach using the CIFAR100 dataset and compare the performance of SINF across popular DNN architectures like VGG16, VGG19, and ResNet50.

**Strengths:**

Novelty in Approach: The paper leverages intrinsic redundancy in latent representations, offering an interesting perspective on accelerating DNNs.

Versatility: The approach is tested across multiple popular DNN architectures, indicating its adaptability.

**Weaknesses:**

However, there are several drawbacks.

## Major

1. There are a lot of existing pruning methods, e.g., DepGraph: Towards Any Structural Pruning
but this paper does not provide any comparison to existing works

2. The acceleration on CIFAR10/CIFAR100 is not satisfied.
For example, DepGraph achieves an 8.92× acceleration with a 3.11 loss in accuracy on CIFAR100 with VGG19.
However, the results provided by the author (~1.72x and 2.83 loss on acc) are worse than previous work.

3. The writing is confusing, it is hard to find the main results.

4. This method applies an additional predictor to discriminate which hyper-class the input should be, which introduces extra inference cost and training effort. Does the provided inference time include the inference time on the additional predictor? Another concern is that the extra network will hurt model generalization ability.

## Minor

1. The authors provided an anonymous GitHub link to share code, but it is an empty link.

**Questions:**

Please refer to my detailed comments in the weakness part.

---

> ### Author Response · Authors · 2023-11-19
>
> # Regarding Weakness 1
>
> > There are a lot of existing pruning methods, e.g., DepGraph: Towards Any Structural Pruning but this paper does not provide any comparison to existing works.**
>
> We thank the reviewer for this comment. Although the end goal of pruning and our proposed approach is to decrease the computational burden of mobile devices related to executing deep neural networks (DNNs), the primary objective of our work is to do so *without the need to fine-tune the DNN*, as in real-world mobile systems the DNN has to be adapted immediately to changing requirements.
>
> The first key reason behind this choice is that fine-tuning DNNs takes a significant amount of time. For example, fine-tuning one of our sub-graphs even for 20 epochs takes on average a minute, which is just for CIFAR-100. For 20 sub-graphs corresponding to 20 semantic clusters, it takes about 20 minutes. In real-world dynamic mobile scenarios where priorities might change, it may be infeasible and energy-expensive to continuously fine-tune the model. For example, if we are deploying a drone for surveillance in a mountain area, that would encounter certain classes (e.g., animals). However, if the UAV operates in an urban area, then it would require a different set of classes (e.g., cars). Given the dynamic nature of the system, allowing fast switching becomes a compelling necessity.
>
> Another reason is that fine-tuning the DNNs for one set of classes can degrade its performance in other tasks due to catastrophic forgetting [1,2]. Conversely, our approach is to *pre-compute and then select at runtime* the sub-graph with the capability of achieving sufficient accuracy on the given set of semantic classes.
>
> DepGraph and many other pruning approaches rely on fine-tuning to improve performance. For this reason, we compare to the most recent work for pruning which adopts same scenario as ours [3] in **Table 1**. We also compare the proposed discriminative capability score (DCS) with other score-based metrics usually used for pruning in **Figure 4**.
>
> [1] Pomponi, J., Scardapane, S. and Uncini, A., 2022. Centroids Matching: an efficient Continual Learning approach operating in the embedding space. Transactions on Machine Learning Research.
> [2] Davari, M., Asadi, N., Mudur, S., Aljundi, R. and Belilovsky, E., 2022. Probing representation forgetting in supervised and unsupervised continual learning. In Proceedings of the IEEE/CVF Conference on Computer Vision and Pattern Recognition (pp. 16712-16721).
> [3] Murti, C., Narshana, T. and Bhattacharyya, C., 2022, September. TVSPrune-Pruning Non-discriminative filters via Total Variation separability of intermediate representations without fine tuning. In The Eleventh International Conference on Learning Representations.
>
> # Regarding Weakness 2
>
> > The acceleration on CIFAR10/CIFAR100 is not satisfied. For example, DepGraph achieves an 8.92× acceleration with a 3.11 loss in accuracy on CIFAR100 with VGG19. However, the results provided by the author (~1.72x and 2.83 loss on acc) are worse than previous work.
>
> Although the review is correct, we point out that conversely from DepGraph and other pruning approaches in literature, our results are obtained *without fine-tuning*. When considering the DCS score for pruning, and enabling fine-tuning, our proposed DCS strategy achieves 9.08x speed up for only 3.27% loss in accuracy which is comparable to DepGraph.
>
> # Regarding Weakness 3
>
> >The writing is confusing, it is hard to find the main results.
>
> We are sorry to know that the reviewer has found the writing to be confusing and main results hard to find. We have updated the manuscript and tried to highlight the main results. We hope that it will make it the results easier to find.

---

> > ### Author Response · Authors · 2023-11-19
> >
> > # Regarding Weakness 4
> >
> > >This method applies an additional predictor to discriminate which hyper-class the input should be, which introduces extra inference cost and training effort. Does the provided inference time include the inference time on the additional predictor?
> >
> > Thank you for giving us the opportunity to clarify this point. The inference time shown in the paper also includes the inference time on the additional predictor. The additional predictor introduces small computational overhead (4.8%) compared to the backbone DNN. This way, it has negligible impact on the inference time. We will clarify this point in the final version of the paper.
> >
> > > Another concern is that the extra network will hurt model generalization ability.
> >
> > We agree that the extra network could hurt model generalization. This is why our proposed approach uses a confidence threshold to avoid this issue. In other words,  if the semantic route predictor is not confident enough, we can bypass the extracted sub-graph and use the base DNN. This decision is made by the feature router. The feature router is presented with the prediction of the semantic route predictor as well as its confidence in its prediction. Provided that the confidence is smaller than the predefined threshold, the feature router activates the full DNN. This way, we can balance the accuracy and latency trade-off. We show in Section 6 (Experimental Evaluation) that with the additional predictor, we can keep the accuracy loss below 0.17%, 3.75%, and 6.75% for VGG19, VGG16, and ResNet50 respectively while reducing inference time by 35%, 29%, and 15% respectively.

---

### Comment · Area_Chair_yMh2 · 2023-12-05
**Final Update**

Dear Reviewers,

Please take this chance to carefully read the rebuttal from the authors and make any final changes if necessary.

Please also respond to the authors that you have read their rebuttal, and give feedback whether their rebuttal have addressed your concerns.

Thank you,

AC

---

### Meta-Review · Area_Chair_yMh2 · 2023-12-15

**Metareview:**

In this paper, the authors address DNN pruning by searching cluster-specific subgraphs for inference, without needing to retrain or fine-tuning. that faster neural network without drastic accuracy loss. The DNN is divided according to the semantic cluster the object belongs to. The discriminative capability score (DCS) is proposed to find the subgraphs and is applied as the pruning criterion. Experiments are conducted on different datasets (CIFAR10/CIFAR100/ImageNet subset) and backbones (VGG16/VGG19/ResNet50), with improved results obtained over IterTVSPrune.

The proposed method in this paper is interesting and novel, and the setting of pruning without needing to fine-tune is a valuable feature. The paper in its current form is not ready for acceptance though. A main weaknesses of the method is the dependency on an external definition of the semantic clusters. The impact of different cluster definitions needs to be carefully studied in the ablation study as part of the paper. Another main weakness is the experiment where the authors focus on relatively old network architectures. There lacks a diversity in the choice of backbones (e.g., CNN vs Transformers or older architectures vs more recent architectures) to validate the generalizability of the proposed approach. Finally, experiments on the complete ImageNet dataset instead of the subset would be much more convincing. This detail is also hidden in the supplementary, which should be changed in future versions.

**Justification For Why Not Higher Score:**

Please kindly refer to weaknesses in the comment section.

**Justification For Why Not Lower Score:**

N/A

---

### Decision · Program_Chairs · 2024-01-16

Reject